# Determinants of Prolonged Length of Hospital Stay in Patients with Severe Acute Ischemic Stroke

**DOI:** 10.3390/jcm11123457

**Published:** 2022-06-16

**Authors:** Kuan-Hung Lin, Huey-Juan Lin, Poh-Shiow Yeh

**Affiliations:** Chi-Mei Medical Center, Department of Neurology, 901 Chung-Hwa Road, Yung-Kang District, Tainan 71004, Taiwan; heartrick@hotmail.com (K.-H.L.); huikuanlin@gmail.com (H.-J.L.)

**Keywords:** prolonged length of stay, length of hospital stay, severe acute ischemic stroke, do-not-resuscitate, functional outcome

## Abstract

Objective: Long hospitalizations are associated with a high comorbidity and considerable hospital cost. Admissions of severe acute ischemic stroke are prone to longer hospitalizations. We aimed to explore the issue and method for improving the length of stay. Methods: From the prospective Stroke Registry between January 2019 and June 2020, acute ischemic strokes with an admission National Institutes of Health Stroke Scale ≥ 15 were identified. Prolonged length-of-stay was defined as in-hospital-stay ≥ 30 days. All clinical characteristics were collected, and all do-not-resuscitate documentations were categorized if the order had been written within 7 days of onset. Results: A total of 212 patients were eligible for severe stroke. Of these, 42 (19.8%) had prolonged length-of-stay and 170 had non-prolonged length-of-stay (median 43 vs. 13 days). The prolonged group was younger, mostly men, and was more likely to be in an independent state and more likely to receive reperfusion therapy, and there was a higher frequency of late do-not-resuscitate orders if signed. Although there was a lower in-hospital mortality rate in the prolonged group (12% vs. 23%), there was a higher proportion with a severe functional state (Modified Rankin Scale = 4–5) among the survivors (97% vs. 87%). Conclusions: Severe acute ischemic stroke patients with a prolonged length-of-stay were younger, mostly male, more likely to receive reperfusion therapy, less likely to have an early do-not-resuscitate order if signed, and more likely to have poor functional status at discharge, although there was a lower rate of in-hospital mortality.

## 1. Introduction

Length of hospital stay (LOS) is often regarded as an indicator of efficiency in hospital management as well as in the quality of care. Prolonged LOS is associated with comorbidities or hospital-acquired infection [1], and it is noted that patients of certain age or with specific diseases are prone to having longer hospital stays [2,3,4,5]. Among the studies investigating the factors contributing to LOS in patients with acute stroke [6,7,8], the most consistent one is stroke severity [9,10]. Severe acute stroke-induced significant disability would result in lengthened hospital stay [11,12], even if high quality acute treatment and professional care in a stroke unit are administered [7,8,13]. For these patients with grave outcomes, it remains to be elucidated whether there is room to improve LOS. 

Do-not-resuscitate (DNR) decisions for patients at the end of life have been promoted to avoid invasive and unsuccessful resuscitation procedures that would lead to a loss of dignity and potentially prolong suffering. For several decades, processes to record DNR decisions have been in place globally, but its applications in acute catastrophic diseases such as stroke are variable, and they may be subject to regional and cultural differences. 

In this study, we aimed to explore the determinants of LOS and particularly the impact of do-not-resuscitate (DNR) orders on LOS by analyzing the clinical characteristics of patients with severe acute ischemic stroke (AIS) from a single hospital-based stroke registry. 

## 2. Methods

The study subjects were assembled from the hospital-based prospective stroke registry at Chi-Mei Medical Center, a teaching hospital in Southern Taiwan with more than 1200 beds. The stroke registry has been approved by the Ethics Committee of Chi-Mei Medical Center, and it conforms to the criteria of the nationwide Taiwan Stroke Registry [14]. In brief, the registry prospectively enrolled patients who had been admitted within 10 days after stroke onset. Patient characteristics, including demographic data, medical history, comorbidities, stroke severity, treatments, hospital course, and complications, were collected according to a pre-defined system. Stroke severity was assessed with the National Institutes of Health Stroke Scale (NIHSS). The functional status at discharge was categorized with the modified Rankin Scale (mRS). Complying with the clinical guidelines for acute stroke management, we started acute medical treatments, including reperfusion therapy, after patients presented to the emergency department and were admitted to the stroke ward or the intensive care unit (ICU), accordingly. 

For the present study, we identified AIS patients with an admission of NIHSS ≥ 15, which we defined as being severe stroke, between January 2019 and June 2020. In-hospital strokes were excluded (Figure 1). We collected the clinical characteristics of interest, including demographic data, history of stroke, vascular risk factors, pre-admission functional state by mRS, pre-admission comorbid condition by the Charlson Comobidity Index (CCI), acute reperfusion therapy (intravenous thrombolytic therapy or/and endovascular thrombectomy), stroke subtypes classified by the Trial of Org 10,172 in Acute Stroke Treatment (TOAST) criteria, surgical intervention, length of ICU stay, functional state at discharge by mRS, and LOS. The cause of in-hospital death was classified as being death from fatal stroke, complications, or other comorbidities. Fatal stroke was defined as mortality associated with clinical evidence of massive brain damage or herniation, which usually occurred within 14 days of stroke onset. Death from complications was defined as mortality that was caused by acute stroke-related complications such as pneumonia or sepsis.

We reviewed medical records to retrieve all DNR documentations, which were signed for the decline of futile resuscitation and ventilator-assisted respiration. Patients with written DNRs were classified as early DNR if they had been documented within 7 days of stroke onset or late DNR if it was more than 7 days. Prolonged LOS was defined as ≥30 days of in-hospital stay, in line with the Monitoring Indicators for Hospital Evaluation from the Ministry of Health and Welfare of Taiwan and the Joint Commission of Taiwan.

### Statistical Analysis

Data were presented as means or medians for numerical variables and proportions for categorical variables. We categorized the study patients into two groups: prolonged LOS (≥30 days) and non-prolonged LOS (<30 days). Clinical characteristics were compared between the groups using the Chi-square test, Fisher’s exact test, Student’s *t*-test, Wilcoxon rank-sum test, Mann–Whitney U test, or Kruskal–Wallis test, where appropriate. We chose the variables with a *p* value < 0.1 on the univariable comparisons and used them as the candidate covariates in multivariable logistic regression to study the potential predictors for prolonged LOS. We used a backward stepwise selection strategy with a P thresthold of 0.05 for the final model. All analyses were performed using STATA Version 14 (StataCorp. LP, College Station, TX, USA). A *p* value of <0.05 (two-sided) was determined to be statistically significant.

## 3. Results

From 1211 patients admitted for AIS during the study period, we identified 212 eligible subjects (17.5%) with a median LOS of 18 days (interquartile range (IQR) 9–28). Among them, 42 patients (19.8%) had prolonged LOS (median LOS 43 days, IQR 35–54) and 170 (80.2%) had non-prolonged LOS (median LOS 13 days, IQR 8–19). The comparison of clinical characteristics between the groups is shown in Table 1. Although the stroke severity demonstrated as admission NIHSS was comparable, the prolonged LOS group was younger, more frequently men, more independent in daily living (mRS ≤ 2) before the admission, and more likely to receive reperfusion therapy and surgical intervention, including decompression or tracheostomy. During the hospital course, patients in the prolonged LOS group were more likely to develop complications such as pneumonia, urinary tract infection, or gastro-intestinal bleeding. DNR orders were present in similar proportions (~40%) for each group, and all were signed by surrogates. However, the timing of DNR documentation was different and more likely to be early in the non-prolonged group and late in the prolonged group (Table 1). 

In-hospital mortality occurred in 12% of the prolonged LOS group and 23% for the non-prolonged group (Figure 2). Among the survivors at discharge, 97% in the prolonged group and 87% in the non-prolonged group were in a severe functional state (mRS = 4–5). Multivariable analyses showed that the statistically significant predictors for prolonged LOS were younger age, male, and receiving reperfusion therapy (Table 2). When analyzing the clinical characteristics between the patients with or without written DNR orders, the patients with early DNR were older and had more severe admissions according to NIHSS and more frequent occurrences of death from fatal stroke than patients with late DNR or without DNR. (Table 3).

## 4. Discussion

Our study demonstrated that, in patients with severe AIS, around one-fifth had prolonged LOS ≥ 30 days. Patients with prolonged LOS had less in-hospital mortality (12% vs. 23%), but most survivors (97% vs. 87%) had poor functional levels at discharge. DNR orders were signed in equal proportions, but they were late (>7 days after stroke onset) in the prolonged LOS group. Younger age, male, and acute reperfusion therapy were the independent risk factors for prolonged LOS.

Many studies have investigated the factors influencing LOS in patients with acute stroke, and they have identified variables including stroke types, stroke severity, stroke-induced disability, the quality of acute stroke care, and even insurance types [6,7,8]. Our study focused on the most significant factor—stroke severity—which was evaluated with admission NIHSS scores in most predictive models [9,10]. Several studies have shown that NIHSS within 6 h after stroke onset was a strong predictor of the clinical outcome after cerebral ischemia [15,16,17,18,19]. We defined admission NIHSS as the baseline NIHSS, because all patients had received appropriate acute treatment at the emergency department. We have analyzed all AIS hospitalizations in the Chi-Mei Medical Center during 2019 and noted that NIHSS scores were positively correlated with LOS. For patients with an NIHSS score of 15 or above, the majority would have a hospital stay that was longer than 8 days. Accordingly, we arbitrarily defined severe AIS as NIHSS of ≥15 in the current study. For such patients with a grave prognosis, clinicians and family also often face the difficult decisions of balancing quality of care and the futility of aggressive treatments. 

In recent years, health care policy has paid attention to the trends of long-hospital-stay because of its implication of considerable costs and resource consumption [2,3]. One study concerning acute stroke care in Taiwan reported that only 10.4% of stroke subjects have prolonged LOS of more than 23 days, but they account for 38.9% of the total person-hospital days and 48.7% of the total in-hospital medical expenses [11]. A better understanding of the determinants of LOS would provide insight for reasonable cost containment without jeopardizing the quality of care. Studies from Western countries have discovered that a younger age and male gender are remarkably associated with prolonged hospitalization [2,3], which occurred more frequently in urban and academic hospitals [3]. Our study demonstrated similar findings in patients with severe AIS. Because aggressive medical managements such as a longer ICU stay and surgical interventions were associated with prolonged LOS, we speculated that societal culture and value preferences might contribute to more proactive medical decision making for severe stroke patients of younger age and male gender, and this would accordingly result in a longer hospitalization. In our hospital, all patients with acute reperfusion treatment were admitted to ICU. Specifically, those receiving endovascular thrombectomy under general anesthesia might acquire more intensive care, which reflects the high expectations from the medical team as well as the family. This might explain our findings that reperfusion therapy was associated with prolonged LOS in this specific group, with high stroke severity. Nonetheless, our analysis also showed that although there was lower in-hospital mortality, and the majority of the severe AIS patients with prolonged LOS had poor functional status at discharge, implying a potential long-term disability burden. 

We noted that do-not-resuscitate orders become generally valued in Taiwan, because life-value issues have gradually been emphasized. In particular, the decision of DNR is not easy in severe AIS [6,10]. Unlike other terminal diseases, it is difficult to predict clinical deterioration and survival for severe AIS because of the lack of consistent or reliable clinical parameters [20,21]. Furthermore, many other factors would influence decision making regarding life-sustaining procedures in these circumstances, such as the pattern, severity, and prognosis of neurological deficits and the burdens of future care [22,23,24]. Previous research has mostly paid attention to the influence of DNR orders regarding the attitude of acute care, the functional outcome, or fatality [22,25]. Our analysis focused on the correlation between DNR orders and prolonged LOS. We discovered that the timing of DNR orders was strongly associated with LOS. With similar proportions (~40%) of DNR, early DNR (signed within 7 days after stroke onset) was less likely in the prolonged LOS group. This may imply that late DNR orders were determined when complex complications or comorbidities continuously evolved or when all possible management efforts were exhausted but to no avail. 

In addition to mortality, the functional status after stroke is also a major outcome of concern for patients and families. Most important, our findings have strongly indicated that, for patients with severe AIS, prolonged LOS saved lives but did not guarantee a favorable functional outcome. In our opinion, the patient’s prognostic information should be conveyed to the family as early as possible, and this discussion should be maintained. Through the process of shared decision making (SDM), the discussion issues should cover the values of the patient and their family, the effectiveness of aggressive interventions, the potential impact of the underlying comorbidities and nosocomial complications from lengthened hospitalization, and finally, the explicit introduction of DNR orders. Above all, early comprehensive discharge planning to set up pragmatic goals such as discharge disposition and rehabilitation for long-term care, as well as the advanced deployment of supported services after discharge, might help alleviate unnecessary prolonged hospitalization [26,27]. 

The strength of our study was that the prospective systematic registry contained important and adequate information for acute stroke management to minimize selection bias and missing data. There are some limitations. First, data from a single hospital-based registry might restrict the generalizability of our findings. Second, as we were limited by the sample size, we were not able to evaluate more detailed predictors for LOS, such as the effectiveness of reperfusion therapy, stroke subtypes, and the socio-economic supports of each patient. Third, the contents of DNR were not specifically analyzed; thus, the impacts of individual items on LOS were unknown. 

## 5. Conclusions

In severe AIS, prolonged LOS was more prevalent among patients of male gender, with a younger age, or who were receiving acute reperfusion therapy. Prolonged LOS, although associated with a lower rate of in-hospital mortality, was often concomitant with a poor functional outcome at discharge. To decrease futile management and to respect the values of patients and their families, explicit prognostic information for patients with severe AIS should be delivered at an early stage and through better shared decision-making processes. In order to improve the length of the hospital stay in such situations, the timely introduction of DNR orders, as well as early preparation for supported discharge services, might be prospects that are worthy of effort by clinicians and stroke care teams.

## Figures and Tables

**Figure 1 jcm-11-03457-f001:**
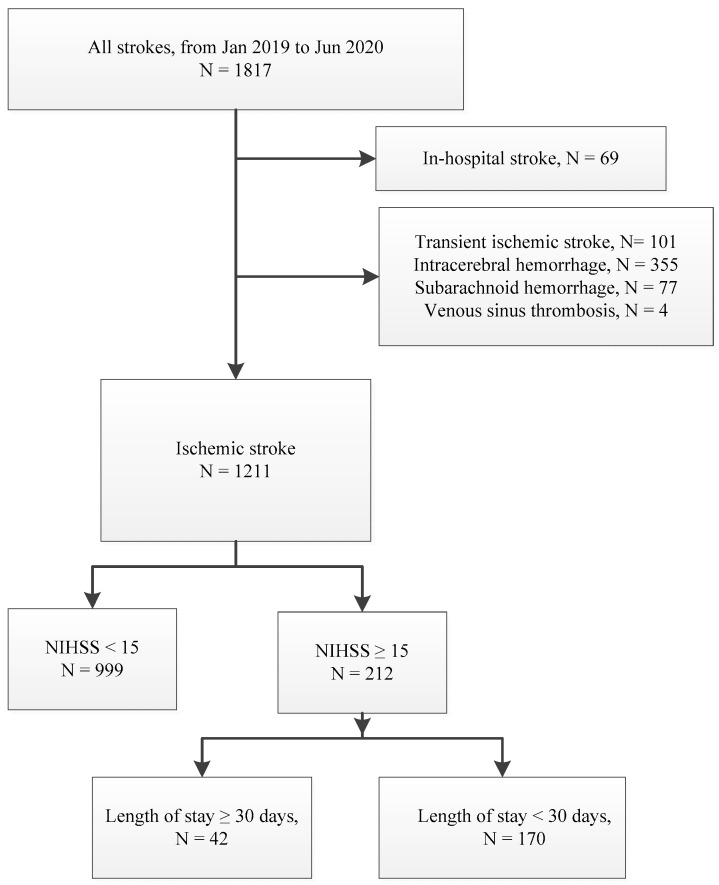
Flow chart of inclusion and exclusion of study patients.

**Figure 2 jcm-11-03457-f002:**
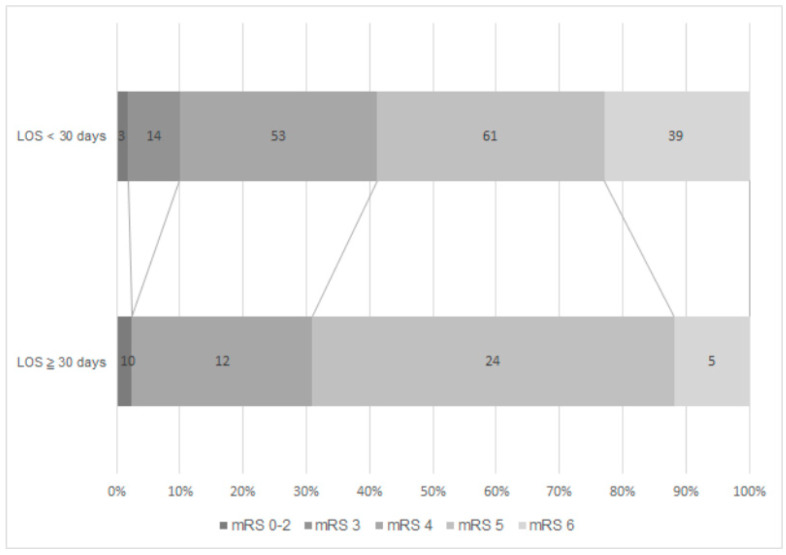
Functional state at discharge by mRS between the prolonged LOS group and the non-prolonged LOS group.

**Table 1 jcm-11-03457-t001:** Demographic data and clinical characteristics * in 212 patients with severe acute ischemic stroke classified by LOS.

	LOS ≥ 30 Days	LOS < 30 Days	*p* Value
	N = 42	N = 170	
Hospital LOS, day, median (IQR)	43 (35–54)	13 (8–19)	<0.01
Median (IQR)			
Age mean ± SD, year	65 ± 17	75 ± 12	<0.01
Male	32 (76%)	79 (46%)	<0.01
Admission NIHSS, median (IQR)	24 (19–30)	23 (19–28)	0.67
History of stroke	13 (31%)	55 (32%)	0.86
Hypertension	21 (50%)	120 (71%)	0.01
Diabetes mellitus	14 (33%)	61 (36%)	0.76
Dyslipidemia	30 (71%)	133 (78%)	0.35
Atrial fibrillation	16 (36%)	51 (30%)	0.31
Ischemic heart disease	6 (14%)	18 (11%)	0.46
Heart failure	2 (5%)	10 (6%)	0.78
Uremia	0	4 (2%)	
PAOD	1 (2%)	1 (0.5%)	0.28
Malignancy	2 (5%)	22 (13%)	0.13
Smoking	7 (17%)	31 (18%)	0.81
Pre-admission CCI,	2.5 (1–5)	2 (1–4)	0.34
median (IQR)			
Pre-admission mRS 0–2	33 (79%)	91 (54%)	<0.01
IV tPA or/and EVT	10 (24%)	18 (11%)	0.02
ICU stay	9 (21%)	19 (11%)	0.08
ICU LOS, day,	14 (13–22)	6 (5–7)	<0.01
median (IQR)			
Surgery	15 (36%)	7 (4%)	<0.01
DNR orders			<0.01
No DNR	25 (60%)	101 (59%)	
Early DNR	4 (10%)	64 (38%)	
Late DNR	13 (31%)	5 (3%)	
Stroke subtypes			
LAA	10 (14%)	54 (32%)	
SVO	0	8 (5%)	
CE	22 (52%)	55 (32%)	
Others	4 (10%)	6 (4%)	
Undetermined	6 (14%)	47 (28%)	
Complications			
Pneumonia	26 (62%)	42 (25%)	<0.001
UTI	7 (17%)	43 (25%)	0.24
UGI bleeding	7 (17%)	7 (4%)	<0.01
Pulmonary edema	0	4 (2%)	
Seizure	1 (2%)	4 (2%)	0.99
Hemorrhagic infarct	5 (12%)	3 (2%)	<0.01
Functional state on discharge mRS			
0–2			0.04
3	1 (2%)	3 (2%)	
4	0	14 (8%)	
5	12 (29%)	53 (31%)	
6 (death)	24 (57%)	61 (36%)	
	5 (12%)	39 (23%)	
Cause of death, No.			0.75
Fatal stroke	3 (7%)	27 (16%)	
Complications	0	4	
Others	2	8	

* Data are No (%), or otherwise specified. Abbreviation: LOS: length of stay; IQR: interquartile range; SD: standard deviation; NIHSS: National Institutes of Health Stroke Scale; PAOD: peripheral arterial occlusive disease; CCI: Charlson comorbidity index; IV tPA: intravenous tissue plasminogen activator; VT: endovascular therapy; ICU: intensive care unit; DNR: do-not-resuscitate; LAA: large-artery atherosclerosis; VO: small-vessel occlusion; CE: cardioembolism; UTI: urinary tract infection; UGI bleeding: upper gastrointestinal bleeding; mRS: modified Rankin Scale.

**Table 2 jcm-11-03457-t002:** Multivariable analysis of predictors for prolonged LOS.

	OR	95% CI	*p* Value
Age (/10 yr increase)	0.68	0.52–0.89	0.004
Male	2.77	1.19–6.48	0.018
Reperfusion therapy	2.90	1.13–7.44	0.027

Abbreviation: OR: odd ratio, CI: Confidence interval.

**Table 3 jcm-11-03457-t003:** Clinical characteristics * in 212 patients with severe acute ischemic stroke classified by DNR orders.

	Early DNR N = 68	Late DNR N = 18	No DNR N = 126	*p* Value
Age mean ± SD, year	79.8 ± 10.4	68.1 ± 12.5	70.3 ± 14.6	<0.01
Male	32 (47%)	12 (67%)	67 (53%)	0.32
Admission NIHSS, median (IQR)	28 (21–33)	25 [19.5–26)	21 (18–25)	<0.01
History of stroke	20 (29%)	7 (39%)	41 (32%)	0.73
Pre-stroke mRS 0–2	32 (47%)	13 (72%)	79 (63%)	0.05
Pre-stroke CCI, median (IQR)	3 (1–5)	2 (1–5)	2 (0–4)	0.03
Discharge mRS 6_death	32 (47%)	3(17%)	9 (7%)	
Cause of death, No.				
Fatal stroke	23 (34%)	1 (6%)	6 (5%)	
Complications	3	0	1	
Others	6	2	2	

* Data are No (%), or otherwise specified. Abbreviation: DNR: do-not-resuscitate; LOS: length of stay; IQR: interquartile range; SD: standard deviation; NIHSS: National Institutes of Health Stroke Scale; mRS: modified Rankin Scale; CCI: Charlson Comorbidity Index; IV tPA: intravenous tissue plasminogen activator; EVT: endovascular therapy; ICU: intensive care unit.

## Data Availability

The datasets used and/or analyzed during the current study are from the stroke registry at Chi-Mei Medical Center. The stroke registry has been approved by the Ethics Committee of Chi-Mei Medical Center.

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
