# Peer review of "Determinants of Prolonged Length of Hospital Stay in Patients with Severe Acute Ischemic Stroke"

_jcm, 2022, doi:10.3390/jcm11123457_

Round 1

Reviewer 1 Report

This study examined the factors associated with the prolonged length of hospital stay in patients with severe acute ischemic stroke. It is a well written study.

Main comments:

1.      Intravenous alteplase therapy is an effective treatment for acute ischemic stroke, but time to treatment is critical. In your study, how many patients arrived hospital by ambulance, and what is the average time between stroke symptom onset and admission? It will be interesting to look at the proportion of arrival by ambulance between LOS30 days and LOS<30 days.

2.      There was a significant difference in receiving IV tPA/EVT between patients with LOS30 days and LOS<30 days (24% vs. 11%). IV alteplase is recommended within 4.5 hours of the time the patient was last known to be well among clinically eligible patients. Were there any contraindications among patients with LOS>30 days that prevented them from IV tPA given? I am surprised with such big difference in IV tPA given between two groups.

3.      Among patients with IV tPA/EVT, how many experienced symptomatic ICH within 36 hours, or life threatening or serious systemic hemorrhage within 36 hours after the treatment?

4.      The proportion of pneumonia was 62% among those with LOS30 days and 25% for LOS<30 days. Any insights for this huge difference? Pneumonia, UTI, and VTE are common complications after acute ischemic stroke. What about VTE in your study population?

Author Response

Reviewer # 1 comments:

  1. Intravenous alteplase therapy is an effective treatment for acute ischemic stroke, but time to treatment is critical. In your study, how many patients arrived hospital by ambulance, and what is the average time between stroke symptom onset and admission? It will be interesting to look at the proportion of arrival by ambulance between LOS≥30 days and LOS<30 days.

Answer 1: Our data revealed that of all the 1211 patients with ischemic stroke and admission via ER, 361 (29.8%) arrived by ambulance. Among the 212 patients with severe ischemic stroke (admission NIHSS ≥ 15), 122 (57.5%) arrived by ambulance. It was true that patients with severe stroke were more likely to utilize ambulance transport. And among the 42 patients with prolonged LOS ≥ 30 days, 28 subjects (66.7%) had the help from the ambulance, comparable with 94 (55.3%) of the 170 patients with LOS < 30 days (p=0.18).

  1. There was a significant difference in receiving IV tPA/EVT between patients with LOS≥30 days and LOS<30 days (24% vs. 11%). IV alteplase is recommended within 4.5 hours of the time the patient was last known to be well among clinically eligible patients. Were there any contraindications among patients with LOS>30 days that prevented them from IV tPA given? I am surprised with such big difference in IV tPA given between two groups.

Answer 2:  For acute stroke managements including acute reperfusion treatment in acute ischemic strokes, we completely followed the guideline from Taiwan Stroke Society. Our clinical performances of acute stroke care were regularly assessed by Taiwan Healthcare Indicator Series (THIS) and Taiwan Clinical Performance Indicators (TCPI). In this study, only one patient was eligible for intravenous tPA but the family refused the treatments. This patient had no prolonged length of hospital stay.

  1. Among patients with IV tPA/EVT, how many experienced symptomatic ICH within 36 hours, or life threatening or serious systemic hemorrhage within 36 hours after the treatment?

Answer 3: Of all 1211 ischemic stroke patients during the study period, 122 patients received iv tPA or/and EVT and 6 patients had symptomatic ICH (6/122 = 4.9%). Among our study patients received iv tPA or/and EVT, 3 patients in the prolonged LOS group and none in the non-prolonged LOS group received surgery for serious hemorrhage.

  1. The proportion of pneumonia was 62% among those with LOS≥30 days and 25% for LOS<30 days. Any insights for this huge difference? Pneumonia, UTI, and VTE are common complications after acute ischemic stroke. What about VTE in your study population?

Answer 4: According to our clinical data, the most common complications were pneumonia, urinary tract infection and upper gastrointestinal bleeding. And no deep vein thrombosis was reported. Symptomatic VTE has been reported less frequent in Asian populations. (J Neurocrit Care 2018; 11(2): 102-109.; Ann Acad Med Singap. 2007 Oct;36(10):815-20.)

Reviewer 2 Report

Thank you for the possibility of reviewing this interesting study regarding the Determinants of Prolonged Length of Hospital Stay in Patients with Severe Acute Ischemic Stroke. 

I found this study very interesting since it reflects our common experience in clinical stroke practice. 

The study is well conceived, and well presented. However I would suggest some points in order to increse its value: 

Even if the article is well written, there are some grammatical mistakes, some typos and some run-on sentences, sometimes certain sentences cut off the momentum of reading, which could possibly be unpleasant for some readers. I would suggest the paper to be corrected by an English speaking person with medical backgroud. 

- In statistical analysis, which is well conceived, can you better explain why did you chose variables with a P value of <0.1 ( and not <0.05, as aspected) in your logistic regression analysis? Additional comment: what kind of logistic regression did you used ? (probit,logit...etc). 

- In results I would probably add a flowchart to better understand patients selection, starting from the group of 1211, to arrive to 212 and the other groups identified.

- In discussion I would like you to add some lines regaring the importance of baseline NIHSS score, there are severl works to which you can get ideas (e.g. Alexandre AM et al. Mechanical thrombectomy in acute ischemic stroke due to large vessel occlusion in the anterior circulation and low baseline National Institute of Health Stroke Scale score: a multicenter retrospective matched analysis. Neurol Sci. 2022 May;43(5):3105-3112. doi: 10.1007/s10072-021-05771-5. Epub 2021 Nov 29. PMID: 34843020). 

- In Discussion section in the two last paragraphs please try to better express the importance of your work.

I think your conclusion lacks content, please try to summarize your study so that readers who read only the conclusion understand your whole work.

- The number of articles cited is, in my opinion, insufficient, please find a list of interesting papers, please consider to add them or some of them in addition to other articles:
- Pilato F et al. Predicting Factors of Functional Outcome in Patients with Acute Ischemic Stroke Admitted to Neuro-Intensive Care Unit-A Prospective Cohort Study. Brain Sci. 2020 Nov 26;10(12):911. doi: 10.3390/brainsci10120911. PMID: 33256264; PMCID: PMC7761293.

- Wirtz MMet al. Predictor of 90-day functional outcome after mechanical thrombectomy for large vessel occlusion stroke: NIHSS score of 10 or less at 24 hours. J Neurosurg. 2019 Dec 20:1-7. doi: 10.3171/2019.10.JNS191991. Epub ahead of print. PMID: 31860816.

- Alexandre AM et al. Posterior Circulation Endovascular Thrombectomy for Large-Vessel Occlusion: Predictors of Favorable Clinical Outcome and Analysis of First-Pass Effect. AJNR Am J Neuroradiol. 2021 May;42(5):896-903. doi: 10.3174/ajnr.A7023. Epub 2021 Mar 4. PMID: 33664106; PMCID: PMC8115369.

Jain M, et al. West J Emerg Med. 2014 May;15(3):267-75. doi: 10.5811/westjem.2013.8.16186. Epub 2014 Apr 15. PMID: 24868303; PMCID: PMC4025522.

- Rincon F, et al. Association between out-of-hospital emergency department transfer and poor hospital outcome in critically ill stroke patients. J Crit Care. 2011 Dec;26(6):620-5. doi: 10.1016/j.jcrc.2011.02.009. Epub 2011 May 18. PMID: 21596517.

Koton S, et al. Derivation and validation of the prolonged length of stay score in acute stroke patients. Neurology. 2010 May 11;74(19):1511-6. doi: 10.1212/WNL.0b013e3181dd4dc5. PMID: 20458067.

Appelros P. Prediction of length of stay for stroke patients. Acta Neurol Scand. 2007 Jul;116(1):15-9. doi: 10.1111/j.1600-0404.2006.00756.x. PMID: 17587250.

Kurtz P et al. Hospital Length of Stay and 30-Day Mortality Prediction in Stroke: A Machine Learning Analysis of 17,000 ICU Admissions in Brazil. Neurocrit Care. 2022 Apr 6. doi: 10.1007/s12028-022-01486-3. Epub ahead of print. PMID: 35381967.

Bösel J, et al. Stroke-related Early Tracheostomy versus Prolonged Orotracheal Intubation in Neurocritical Care Trial (SETPOINT): a randomized pilot trial. Stroke. 2013 Jan;44(1):21-8. doi: 10.1161/STROKEAHA.112.669895. Epub 2012 Nov 29. PMID: 23204058.

Siepmann T, at al. The Effects of Pretreatment versus De Novo Treatment with Selective Serotonin Reuptake Inhibitors on Short-term Outcome after Acute Ischemic Stroke. J Stroke Cerebrovasc Dis. 2015 Aug;24(8):1886-92. doi: 10.1016/j.jstrokecerebrovasdis.2015.04.033. Epub 2015 Jun 19. PMID: 26099557.

Round 2

Reviewer 1 Report

The authors addressed my comments.